# Osmanthus-Loaded PVP/PVA Hydrogel Inhibits the Proliferation and Migration of Oral Squamous Cell Carcinoma Cells CAL-27

**DOI:** 10.3390/polym14245399

**Published:** 2022-12-09

**Authors:** Bin Huang, Chizhou Wu, Yuzhu Hu, Lu Rao, Mingzhe Yang, Mengyao Zhao, Huangqin Chen, Yuesheng Li

**Affiliations:** 1Department of Stomatology, School of Stomatology and Ophthalmology, Xianing Medical College, Hubei University of Science and Technology, Xianning 437100, China; 2Hubei Key Laboratory of Radiation Chemistry and Functional Materials, Non-Power Nuclear Technology Collaborative Innovation Center, Hubei University of Science and Technology, Xianning 437100, China

**Keywords:** osmanthus, PVP/PVA, hydrogel, electron beam radiation, migration

## Abstract

Conventional medical agents for oral squamous cell carcinoma (OSCC) with some adverse effects no longer meet the needs of the public. In this study, the prognosis-related hub genes of osmanthus-targeted therapy for OSCC were predicted and analyzed by network pharmacology and molecular docking. Osmanthus was extracted using the ethanol reflux method and osmanthus-loaded PVP/PVA (OF/PVP/PVA) hydrogel was prepared by electron beam radiation. The molecular structure, crystal structure and microscopic morphology of hydrogels were observed by Fourier transform infrared spectroscopy (FTIR), X-ray diffraction (XRD) and scanning electron microscopy (SEM), respectively. OSCC cells CAL-27 were cultured with OF/PVP/PVA hydrogel at different concentrations of extract to discover cell proliferation by MTT assay. The scratching test and JC-1 staining were used to observe the migration and mitochondrial membrane potential. Through experimental exploration, we found that a total of six prognosis-related targets were predicted, which are *PYGL*, *AURKA*, *SQLE,* etc., and osmanthus extract had good binding activity to *AURKA.* In vitro, except for proliferation inhibition, OF/PVP/PVA hydrogel prevented cell migration and changed the mitochondrial membrane potential of CAL-27 cells at a concentration equal to or greater than 50 μg/mL (*p* < 0.05). The addition of autophagy inhibitor chloroquine and 3-methyladenine weakened the migration inhibition of hydrogel (*p* < 0.05).

## 1. Introduction

Oral squamous cell carcinoma (OSCC) is a malignant tumor that occurs in the lip, tongue, salivary gland, gingiva, floor of the mouth, oropharynx, buccal mucosa and other parts of the mouth. OSCC, accounting for more than 90% of oral malignant tumors, is highly malignant and easy to transfer, which seriously endangers people’s physical and mental health [1,2,3]. The five-year survival rate of OSCC is only 50–60% [4] and the mortality rate increases by about 2% per year [5]. Each year, more than 300,000 new cases occur and there are more than 140,000 deaths from OSCC worldwide [6]. Multimodal therapy, including surgery, chemotherapy, biotherapy and radiotherapy, is the main treatment for OSCC patients at present, but these methods may have some adverse effects on patients [7]. Therefore, the search for herbal medicines with OSCC therapeutic effects has become a research hotspot.

Osmanthus is one of China’s top ten traditional flowers, with a cultivation history of more than 2500 years. Xianning City, Hubei Province, as one of the historically famous “five major osmanthus producing areas”, has been named the “The hometown of osmanthus in China” due to its large area, complete variety, high yield, good flower quality and many ancient laurels [8,9]. According to the *Compendium of Materia Medica* and *Jiangsu Materia Medica*, osmanthus seeds have hepatoprotective and stomach-protective properties [10,11]. The roots of osmanthus have therapeutic effects on inflammation, cardiovascular diseases, cancer and rheumatoid arthritis [12]. The flowers of osmanthus are also used as a natural antioxidant, regulating peroxidation reactions and controlling the production of free radicals in the body [13]. Reactive oxygen species (ROS) target several major signaling molecules involved in cancer cell migration and invasion, including kinases and transcription factors. Cancer cells exhibiting high ROS production have higher migration and invasive behavior [14]. The inhibition of ROS with antioxidants attenuates the migration and invasion ability of cancer cells by decreasing the ROS level. The anticancer efficacy of natural flavonoids is largely attributed to their powerful antioxidant capacity and anti-inflammatory activity, which are closely related to cancer cell survival, proliferation, differentiation, migration, angiogenesis, etc. [15]. Although the osmanthus fragrans flower has shown good anti-inflammatory and anti-tumor effects [16,17], its effects on OSCC proliferation and migration are rarely reported.

Currently, compared with traditional tumor treatment strategies, drug-loaded hydrogel as a drug reservoir system has attracted wide attention due to its on-demand drug release and deep tissue penetration. This not only enhances therapeutic effects but also significantly overcomes severe side effects and drug resistance against various kinds of tumors [18]. Polyvinyl pyrrolidone (PVP) is a biocompatible, temperature-resistant polymer obtained by free-radical polymerization of the monomer N-vinylpyrrolidone, which exhibits complex affinities for both lipophilic and hydrophilic drugs. The films that are obtained from PVP have a shiny appearance, and when dried they are quite translucent and resistant [19]. PVP is commonly used as a matrix or an additive for the controlled release of drugs through coprecipitation of other drugs due to its important properties, such as low toxicity, biodegradability, thermal and chemical resistance, ability to complex with hydrophilic and hydrophobic molecules and solubility in water and organic solvents [20]. Polyvinyl alcohol (PVA) is known as a film-forming agent, used for increasing the mechanical properties of films and also to obtain blends with other hydrophilic polymers [21]. It is widely used to prepare drug-loaded hydrogels by chemical cross-linking with the water-soluble polymer PVA [22]. The PVP/PVA composite nanofiber membrane made by electrospinning technology not only improves the drug release curve but also shows good mechanical properties, thermal stability and biocompatibility [23]. Recently, electron beam irradiation in biological materials synthesis has received attention [24]. This method utilizes direct electrons to generate radical reactions, and the penetration depends on the electron energy (usually in MeV or KeV), without any initiators, solvents or chemicals for copolymerization. Therefore, we want to use electron beam irradiation to prepare osmanthus extract-loaded PVP/PVA hydrogels for the treatment of OSCC.

However, like all herbal medicines, osmanthus extracts are multi-component, multi-target and multi-pathway, which are difficult to analyze systematically and comprehensively by traditional experimental methods. In recent years, network pharmacology has been increasingly used in herbal medicine for disease treatment [25]. Network pharmacology integrates the methods of systems biology, multidirectional pharmacology and computer technology to study the relationship between drugs, targets and diseases; visualize drug components; target genes for disease treatment, and protein interaction networks; and study the mechanism of action of drugs for disease treatment from a holistic perspective; which coincides with the research idea of a multi-component, multi-target and multi-pathway approach to traditional Chinese medicine [26]. Therefore, this study intends to explore the complex relationship between osmanthus and OSCC by combining network pharmacology, molecular docking methods and experimental validation. This will provide an experimental basis for future research on the pharmacological effects of osmanthus and reveal the mechanism of osmanthus in the treatment of OSCC.

## 2. Materials and Methods

### 2.1. Data Download and Acquisition of Potential Targets

The mRNA expression profiles and corresponding clinical information of OSCC were downloaded from the TCGA database (https://portal.gdc.cancer.gov, accessed on 10 September 2022), including 44 normal samples and 504 OSCC cancer samples, and the differential expression of mRNA was investigated using the Limma package of R software (version:3.40.2). The main components of osmanthus were found by searching the literature, and their 2D structures were downloaded from PubChem (https://pubchem.ncbi.nlm.nih.gov, accessed on 10 September 2022) and imported into SwissTargetPrediction (http://www.swisstargetprediction.ch, accessed on 10 September 2022) to predict potential therapeutic targets. Venn diagrams were plotted using the Venny 2.1.0 online database (http://bioinfogp.cnb.csis.es/tools/venny/index.html, accessed on 10 September 2022) to obtain potential targets for osmanthus treatment with OSCC.

### 2.2. Gene Ontology (GO) Enrichment and Kyoto Encyclopedia of Genes and Genomes (KEGG) Pathway Analysis

GO analysis and KEGG pathway enrichment analysis were performed on the overlapping genes using R language cluster Profiler package and the difference was statistically significant at *p* < 0.05. The obtained biological progress (BP), cellular components (CC), molecular function (MF) and KEGG pathway data were analyzed to discover the biological functions among these genes. The first 10 entries were also visualized and plotted as chord plot. A drug–component–target–pathway analysis network diagram was constructed using Cytoscape (version 3.8.0) software.

### 2.3. Screening for Hub Genes and Prognosis-Related Hub Genes

The STRING online database (https://string-db.org, accessed on 10 September 2022) was used to construct the protein–protein interaction (PPI) network of overlapping genes, and the parameters were set as follows: species “Homo sapiens “, the minimum interaction threshold “medium confidence (0.400)”, and the results were imported into Cytoscape (version 3.8.0) for visualization and analysis. The hub genes were obtained using the MCC algorithm of the Hubba plugin.

Univariate COX logistic regression analysis was performed by the R software survival package to find genes associated with OSCC prognosis, genes with higher final risk scores were screened by multivariate COX logistic regression analysis, and hub genes with prognostic effect of osmanthus on OSCC were screened by intersection with hub genes.

### 2.4. Molecular Docking

The 3D structure of the main components of osmanthus extract was downloaded using PubChem database, the target protein structure of the main components of osmanthus extract for OSCC was downloaded from the RCSB PDB (http://rcsb.org, accessed on 10 September 2022) database, the solvent molecules and ligands were removed using Pymol software, and after hydrogenation and electron addition using AutoDock vina software, molecular docking was performed and free binding energy was obtained.

### 2.5. Extraction of Osmanthus Extract and Preparation of Osmanthus Extract-Loaded PVP/PVA (OF/PVP/PVA) Hydrogel

The fresh osmanthus was dried at 80 °C immediately after picking and crushed by a traditional Chinese medicine pulverizer. The method of ethanol reflux was used for extraction. The ratio of osmanthus and 80% ethanol solution was 1:40 (g/mL), the reflux extraction time was 3 h, and the extraction temperature was 70 °C. The obtained extract was concentrated by rotary evaporation of ethanol at 65 °C in a rotary evaporator and filtered through a 0.45 μm microporous membrane to obtain osmanthus extract.

Ten percent PVP (PVP K30, Mw: 44,000–54,000, Sinopharm Chemical Reagent Co. Ltd., Shanghai, China) and 8% PVA (Mw: 1750 ± 50, Sinopharm Chemical Reagent Co. Ltd., Shanghai, China) aqueous solution were mixed at a ratio of 1:2, and osmanthus extract was added into the mixture to make final concentrations of 400, 200, 100, 50, 25 μg/mL, respectively. At room temperature, a 1 MeV electron beam accelerator was used to emit electron beam irradiation with a total irradiation dose of 30 kGy and a dose rate of 10 kGy/passr (1 MeV, Wasik Associates, Dracut, MA, USA).

### 2.6. Characterization

The prepared hydrogel samples were dried and pulverized with a universal pulverizer to obtain sufficiently fine powder. The sample was uniformly mixed with potassium bromide powder in a certain proportion, pressed into tablets, and placed in Fourier transform infrared spectroscopy (FTIR) (NICOLET 5700 spectrometer, Thermo Fisher Nicolet, America) to measure the absorption peak in the range of 4000–400 cm^−1^. The crystalline or amorphous nature of OF/PVP/PVA hydrogel was evaluated by X-ray diffraction (XRD) analysis (DMAX-D8X, Rigaku, Japan), within the range of 10–80° with an angle of 2θ 2°/min. The microstructure of the hydrogel was observed after freeze-drying by scanning electron microscopy (SEM) (S-4800 scan electron microscopy, Hitachi, Japan). The stability of hydrogel in deionized water was determined by a UV-visible spectrophotometer (UV).

The swelling ratio of hydrogel was conducted at 37 ± 1 °C in distilled water. The samples were processed to similar shape and weighed before soaked in distilled water. After a period of time, the samples were taken out and the water remaining on the surface of the samples was removed by filter paper. The swelling ratio (SR) of a sample was calculated according to Equation (1):(1)$$SR=\left(\frac{W_{s}-W_{d}}{W_{d}}\right)\times 100\%$$
where SR is the swelling rate when hydrogel reaches saturation, %; *Ws* is the weight of hydrogel when it reaches swelling equilibrium in distilled water, g; *W_d_* is the weight of hydrogel when drying, g.

For the gel fraction test, the identically shaped samples were dried until the weight of the drying sample reached a constant value (*W_d_*). Then, the dried samples were immersed in 80 °C constant temperature water bath for 24 h. Finally, the samples were dried and weighed (*W_e_*). The gel fraction was calculated according to Equation (2):(2)$$F_{G}=\frac{We}{Wd}\times 100\%$$

### 2.7. Cell Culture and MTT Assay for Cell Activity

CAL-27 cells (Procell Life Science & Technology Co. Ltd., Wuhan, China) were incubated in DMEM medium (GIBCO, Invitrogen Corporation, New York, NY, USA) containing 10% fetal bovine serum (Hyclone, Logan, UT, USA), 100 U/mL penicillin and 100 mg/mL streptomycin at 37 °C, 5% CO_2_, 95% relative humidity (Thermo Forma 3111, Waltham, MA, USA) and cells in logarithmic growth phase were taken for the experiment.

CAL-27 cells were inoculated in a 96-well plate and incubated OF/PVP/PVA hydrogels with different concentrations of osmanthus extract for 24 h, MTT assay was used to detect the OD value of each well and calculate the cell viability (Microplate reader, Bio-Tek, VT, USA).

### 2.8. Cell Migration Ability Detected by Scratching Assay

After CAL-27 cells were seeded on 6-well plates, scratches were made with the tip of a 200 μL pipette and photographed under a microscope (Olympus, Japan). Then OF/PVP/PVA hydrogels with different concentrations of osmanthus extract were added and incubated with cells for 24 h, the area of the scratches was measured by Image J software and cell migration rate was analyzed.
Cell migration rate = (0 h scratch area − scratch area after culture)/0 h scratch area × 100%

### 2.9. Detection of Mitochondrial Membrane Potential Changes by JC-1 Staining

The cells were inoculated in a 6-well plate and treated with OF/PVP/PVA or PVP/PVA hydrogels for 24 h. One milliliter of JC-1 staining solution and 1 mL of culture medium were added, mixed thoroughly and incubated for 20 min at 37 °C in a cell incubator. The change of mitochondrial potential was measured by the relative change of red and green fluorescence (Olympus, Japan).

## 3. Results

### 3.1. Potential Targets of Osmanthus Extract for OSCC Treatment

Osmanthus extract is a complex traditional medicine with multi-component and multi-target characteristics. Its medicinal value and molecular mechanism for OSCC needs to be further explored. Network pharmacology has been used to predict drug–disease interaction networks in different research fields, such as the discovery of new drugs, the elaboration of pharmacological mechanisms and the exploration of new targets [21,22]. In this study, a total of 2115 differential genes of OSCC were found, including 1376 upregulated and 739 downregulated genes (Appendix A). A literature search revealed that the following seven major components of osmanthus extract were identified: quercetin, rutin, genistin, isorhamnetin, kaempferol, naringin and verbascoside. The components were imported into SwissTargetPrediction and a total of 275 potential therapeutic targets were obtained after combining and de-duplicating. Using the Venny online website to draw Venn diagrams, 54 overlapping genes of OSCC differential genes and osmanthus extract potential targets were obtained (Appendix A).

### 3.2. Functional Enrichment Analysis of Potential Targets

GO analysis showed that the overlapping genes were mainly associated with response to oxygen-containing compounds, response to oxidative stress, extracellular matrix disassembly and other biological processes (Appendix A).

Concerning cellular components, the overlapping genes were enriched in microtubule cytoskeleton, extracellular matrix, extracellular region part, etc. (Appendix A).

With regards to molecular functions, they are mainly in catalytic activity, protein kinase activity, phosphotransferase activity, etc. (Appendix A). Pathways enriched by KEGG were closely related to p53 signaling pathway, nitrogen metabolism, etc. (Appendix A). The overlapping targets and KEGG pathways were constructed as a network map, as shown in Appendix A. The network map contains 7 main components of osmanthus extract, 54 overlapping targets, and all KEGG signaling pathways.

### 3.3. Hub Genes and Prognosis-Related Hub Genes of Osmanthus Targeted Therapy for OSCC

A PPI network of 54 overlapping genes was constructed using the STRING database (Appendix A) and the top ten genes (hub genes) obtained were as follows: *CDK6*, *AURKA*, *CDK1*, *TYMS*, *TOP2A*, *AURKB*, *CDK2*, *PLK1*, *CHEK1*, *KIF11*

Cox proportional hazards regression model is a semiparametric regression model that takes survival time and survival status as dependent variables and can simultaneously analyze the impact of various factors on survival. In this study, univariate Cox regression analysis of the transcriptome data of OSCC samples in TCGA database showed that 1713 genes were associated with the prognosis of OSCC patients and only 9 prognostic genes remained after multivariate Cox regression analysis. A Cox proportional hazard regression model consisting of *PYGL*, *AURKA*, *SQLE*, *LGALS1*, *CES1* and *APP* was constructed by intersecting 9 prognostic genes with 54 overlapping genes.
Risk score = (0.1138)*AURKA + (0.0729)*LGALS1+ (0.0411)*APP + (0.1025)*SQLE + (0.123)*PYGL + (0.0499)*CES1.

*AURKA* was found to be a hub gene for the targeted treatment of OSCC and have a good prognostic effect, which is similar to the prediction results of prognostic biomarkers for OSCC by other scholars [27,28]. Therefore, *AURKA* plays a valuable role in the process of OSCC and targeting *AURKA* is a promising adjunctive therapy.

### 3.4. Binding Activity of Osmanthus Extract and Prognosis-Related Hub Gene AURKA

To verify the targeted therapeutic effect of osmanthus extract on *AURKA* in OSCC, seven main components of osmanthus extract and the key prognosis-related target protein *AURKA* were analyzed using AutodockTools software (Appendix A). The results showed that the average binding energy of seven main components of osmanthus extract with AURKA was −7.69 ± 0.46 kcal/mol. Among them, the absolute value of binding energy of acteoside is the largest (−8.28 kcal/mol), while that of kaempferol is the smallest (−6.96 kcal/mol). It is generally believed that when the ligand binds to the receptor protein in a stable conformation, the lower the energy, the more likely the interaction is to occur. When the absolute value of binding energy is >4.25 kcal/mol, it indicates that it has certain activity, >5.0 kcal/mol indicates that it has good binding activity, and >7.0 kcal/mol indicates that it has strong binding activity. Among seven main components, the absolute value of binding energy between *AURKA* and osmanthus extract was greater than 7 kcal/mL except kaempferol, indicating that there was a strong relationship between them, and osmanthus extract could exert therapeutic effects by targeted binding of *AURKA*.

### 3.5. Molecular Structure, Crystal Structure and Microscopic Morphology of OF/PVP/PVA Hydrogel

Osmanthus was extracted using the ethanol reflux method and the main components of the extract were flavonoids tested by UV-Vis spectrophotometer in previous studies [29]. The OF/PVP/PVA hydrogel was successfully prepared and the loading rate (wt%) was 40%, 20%, 10%, 5%, 2.5% and 0%, respectively. Molecular structure, crystal structure and microscopic morphology of hydrogels were observed by FTIR, XRD and SEM, respectively.

The absorption peaks of PVA at 2941 cm^−1^ and 1096 cm^−1^ are caused by stretching vibration of –CH_2_– and –CO– groups, respectively, while the broad peak at 3432 cm^−1^ is the characteristic spectrum peak of unbonded -OH groups. The characteristic absorption peaks of PVP at 2950 cm^−1^, 1677 cm^−1^ and 3448 cm^−1^ are caused by stretching vibration of -CH_2_-, –C=O– and –NH–amide groups, respectively. The absorption peak at 1288 cm^−1^ is caused by the bending vibration of -NH-amide group. The absorption peak at 1461 cm^−1^ is caused by the bending deformation of –CH_2_– group. The infrared spectrogram of OF/PVP/PVA is basically the superposition of the spectrograms of PVA and PVP. It can be seen from the Figure 1 that the PVA hydroxyl peak of 3432 cm^−1^ has been shifted and its intensity has also changed, indicating that the carbonyl group in PVP interacts with the hydroxyl group of PVA affecting the intermolecular hydrogen bond binding of PVA hydrogel. The absorption peak near 849 cm^−1^ is the characteristic absorption peak of PVA, indicating that the introduction of osmanthus does not destroy the structure of PVP/PVA polymer.

Figure 2 shows the XRD patterns of PVP, PVA, osmanthus extract and OF/PVP/PVA hydrogels with different concentrations of osmanthus extract, respectively. It can be seen from the figure that osmanthus extract has a broad diffraction peak at 15–30°, and PVA is a highly crystalline polymer with a strong diffraction absorption peak at 2θ = 19.6°. After PVP, PVA and osmanthus extracts were cross-linked by radiation, the peak of PVA became weaker, which changed the original crystal structure of PVA and reduced crystallinity.

Figure 3A,B are scanning electron microscopy images of PVP/PVA and OF/PVP/PVA hydrogel sections enlarged by 200× and 1000×, respectively. As can be seen from the figure, both hydrogels are relatively flat and uniform, without defects, and there is no obvious phase separation region, which indicates that PVA and PVP have good compatibility, and the addition of osmanthus extract does not affect the density of hydrogels.

Figure 4 and Figure 5 show the stability and swelling rate of PVP/PVA hydrogel and OF/PVP/PVA hydrogel with the concentration of 100 μg/mL, respectively. It can be seen from the figure that the hydrogel has good stability, and the addition of osmanthus extract has no significant effect on the stability and swelling rate of the hydrogel. The gel fraction of hydrogels with different concentrations of osmanthus extract was also detected, which were 73.0%, 70.5%, 77.7%, 75.2%, 75.1% and 70.7%, respectively (the concentration of osmanthus extract ranged from 400 μg/mL to 0 μg/mL). The results showed that the addition of osmanthus extract had little effect on swelling rate.

### 3.6. Inhibitory Effect of OF/PVP/PVA Hydrogels on CAL-27 Proliferation

The uncontrolled proliferation and impaired cell cycle of cancer cells often leads ab-normal cells to invade and metastasize to other parts of the body [30]. Mitochondrial dysfunction-induced cell-death signaling and ROS generation correlate with the malignancy and invasiveness of cancer cells [31]. A number of experimental systems and numerous signaling molecules have been reported to be involved in tumor cell proliferation, such as ERK/MAPK, NF-κB and JNK et al. [14]. Flavonoids have exhibited dual action regarding ROS homeostasis. Under normal conditions, they act as antioxidants, while in cancer cells they are potent pro-oxidants triggering the apoptotic pathways and suppressing cancer cell proliferation [32].

In this study, the hydrogels loaded with different concentrations of osmanthus extract were co-cultured with CAL-27 cells for 24 h, and cell proliferation was detected by MTT assay. The results (Figure 6) showed that compared with the control group, when the concentration of osmanthus extract was greater than or equal to 400 μg/mL, the cell proliferation decreased significantly with the increase in extract concentration (*p* < 0.05), showing strong cytotoxicity in a concentration-dependent manner. When the concentration of osmanthus extract in the hydrogel is equal to or less than 400 ug/mL, the hydrogel is not cytotoxic to CAL-27.

### 3.7. Inhibitory Effect of OF/PVP/PVA Hydrogels on CAL-27 Migration

The cell scratch test is a simple method to determine the migration and repair ability of cells, which simulates the migration process of cells in the healing process in vivo. When the cells have grown to the point where they have fused into a single layer, a blank area, called a scratch, is artificially created on the fused monolayer. The cells at the edge of the scratch gradually move into the blank area to heal the “scratch”. Images were captured at the beginning and at regular intervals during cell migration, and the speed of cell migration was quantitatively compared by comparing pixels in the scratched area of the image. As shown in Figure 7, the area of scratches in each group before cell migration was about 714,971 ± 8869 pixel^2^, with no significant difference. After co-culturing with OF/PVP/PVA hydrogels with different concentrations of osmanthus extract for 24 h, the scratch area decreased significantly in a concentration dependent manner. The largest area was 634,344 ± 12,563 pixel^2^ in the 400 μg/mL concentration group, and the smallest area was 404,377 ± 10,721 pixel^2^ in the control group. Cell migration rate also decreased from 44% to 12%. Except for the 25 μg/mL group, the cell migration rate of the OF/PVP/PVA hydrogel group was lower than that of the control group, which perfectly demonstrated the blocking effect of osmanthus extract on the migration ability of CAL-27.

### 3.8. Effect of OF/PVP/PVA Hydrogels on Mitochondrial Membrane Potential

JC-1, also known as CBIC2(3), is an ideal fluorescent probe widely used to detect mitochondrial membrane potential ΔψM with the molecular formula C25H27Cl4IN4. The JC-1 fluorescent probe showed potential dependent accumulation in mitochondria. When the mitochondrial membrane potential is high, the JC-1 fluorescent probe aggregates in the mitochondrial matrix to form J-aggregates, which can produce red fluorescence. When the mitochondrial membrane potential is low, JC-1 cannot aggregate in the mitochondrial matrix, and can only exist in the form of a monomer, which can produce green fluorescence. The decrease in membrane potential can be easily detected by the transition of JC-1 from red to green fluorescence, and this transition can also be regarded as a hallmark event in the early stage of apoptosis.

In this study, an enhanced mitochondrial membrane potential assay kit with JC-1 was used to detect the effect of OF/PVP/PVA hydrogels containing different concentrations of osmanthus extract on mitochondrial membrane potential of CAL-27. The results (Figure 8) showed that a lot of red fluorescence and little green fluorescence were observed in the mitochondria of control cells. There was no significant difference between the 25 μg/mL group and the control group. When the concentration of osmanthus extract in OF/PVP/PVA hydrogel was greater than or equal to 50 μg/mL, the green fluorescence in mitochondria increased significantly. These results indicated that OF/PVP/PVA hydrogel promoted CAL-27 cell apoptosis to target OSCC, which was consistent with the previous results of network pharmacological analysis.

### 3.9. Effects of an Autophagy Inhibitors on OF/PVP/PVA Hydrogel Inhibition of Cal-27 Cell Migration and Downregulation of Mitochondrial Membrane Potential

GO enrichment analysis in network pharmacology suggested that targeted treatment of OSCC by osmanthus extract was closely related to response to oxidative stress. When a large amount of ROS is produced, LC3 on the phagosome is recruited to modify the phagosome and induce autophagy. Due to the apparent close link between autophagy and cancer, which is indeed a critical role, the use of autophagy as a target for cancer therapy has become a focus of attention [33]. In this study, JC-1 immunofluorescence staining results showed that OF/PVP/PVA hydrogels containing different concentrations of osmanthus extract could cause the decrease of mitochondrial membrane potential of CAL-27, suggesting that hydrogels could induce autophagy. To explore the role of autophagy in hydrogel-targeted OSCC treatment, the autophagy inhibitors chloroquine (CQ) and 3-methyladenine (3-MA) were used for pretreatment, and the cell migration and mitochondrial membrane potential of CAL-27 were observed (Figure 9). The results of immunofluorescence staining showed that there was little or no green fluorescence dot cluster in the cells of the control group. After the OF/PVP/PVA hydrogel containing 200 μg/mL osmanthus extract was added for 24 h, a large number of dot clusters of green fluorescence appeared in the cytoplasm. This indicated that JC-1 was a monomer, and the mitochondrial membrane potential was low at this time, and autophagy had occurred in the cells. After combined treatment of hydrogel and 1 μM CQ, the intracellular dot clusters of green fluorescence were further accumulated because CQ can block the degradation of autophagosomes at the lysosomal level but cannot inhibit the formation of autophagosomes. When combined with 3-MA, the intracellular green fluorescence was significantly decreased, which was significantly lower than that of the OF/PVP/PVA hydrogel-alone treatment group. This is because 3-MA can inhibit the formation of autophagosomes and prevent the occurrence of autophagy. Therefore, the mitochondrial membrane potential is high, and JC-1 aggregates in the matrix of the mitochondria, forming a polymer (J-aggregates) that produces red fluorescence. This also shows from another aspect that the increase of autophagosomes in CAL-27 cells under the action of OF/PVP/PVA hydrogel is indeed the result of the activation of autophagy mechanism, rather than an illusion caused by the inhibition of the autophagy degradation pathway. This study also observed the effect of autophagy inhibitors on the ability of OF/PVP/PVA hydrogel to attenuate CAL-27 migration. The results showed that the combination of CQ or 3-MA with hydrogel could effectively block the inhibitory effect of hydrogel on CAL-27 migration activity, which further indicated that OF/PVP/PVA hydrogel may exert its inhibitory effect by activating autophagy of CAL-27 cells.

## 4. Conclusions

A total of six prognosis-related hub genes of osmanthus-targeted therapy for OSCC, such as PYGL, AURKA, SQLE, were predicted by bioinformatics analysis and network pharmacology. *AURKA* was selected for molecular docking analysis with seven main components of osmanthus extract, and all the components exerted strong binding activity. Osmanthus was extracted using the ethanol reflux method and OF/PVP/PVA hydrogel was prepared by electron beam radiation. Molecular structure, crystal structure and microscopic morphology of hydrogels were observed by FTIR, XRD and SEM, respectively. The infrared spectrogram of OF/PVP/PVA is basically the superposition of the spectrograms of PVA and PVP. The introduction of osmanthus does not destroy the structure and the density of PVP/PVA polymer but reduces its crystallinity. The cell proliferation, migration and mitochondrial membrane potential of CAL-27 were also studied to confirm the inhibition of OF/PVP/PVA hydrogel and explore its mechanism, which involved in cell apoptosis. To explore the role of autophagy in hydrogel-targeted OSCC treatment, the autophagy inhibitors CQ and 3-MA were used for pretreatment, and the cell migration and mitochondrial membrane potential of CAL-27 were observed. The results showed that OF/PVP/PVA hydrogel exerted its inhibitory effect by activating autophagy of CAL-27 cells. This has very important scientific significance for OF/PVP/PVA hydrogels for anti-tumor procedures, which will enrich and complement the application of hydrogel in the field of natural drug delivery.

Recently, smart hydrogels, responding intelligently to changes in the environment, have been receiving increasing interest [34]. They have been widely used in cancer research for highly targeted delivery and on-demand release. In the future, precisely tunable and highly controlled hydrogels will be employed, and researchers should make joint efforts to ensure that the hydrogel delivery system is secure and effective.

## Figures and Tables

**Figure 1 polymers-14-05399-f001:**
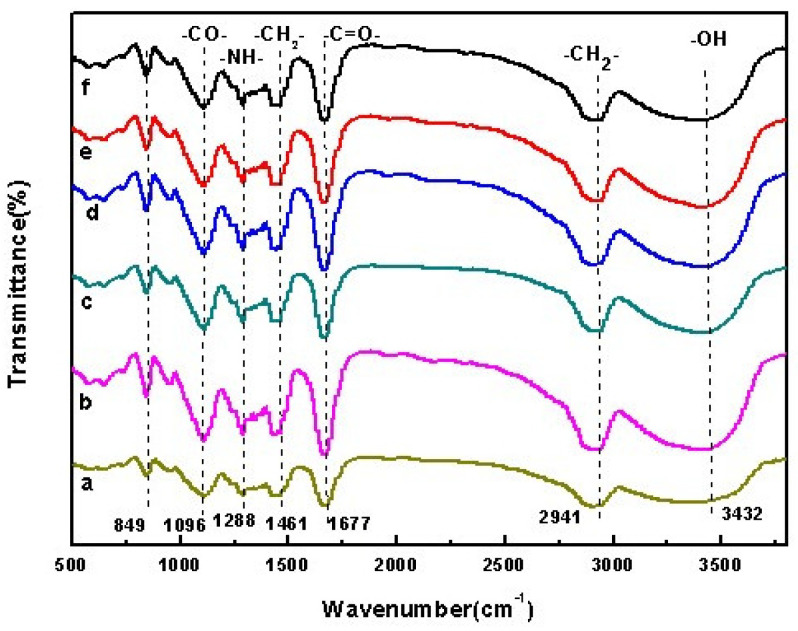
FTIR absorption spectrum of OF/PVP/PVA hydrogels with different concentrations of osmanthus extract. (**a**): 400 μg/mL; (**b**): 200 μg/mL; (**c**): 100 μg/mL; (**d**): 50 μg/mL; (**e**): 25 μg/mL; (**f**): 0 μg/mL.

**Figure 2 polymers-14-05399-f002:**
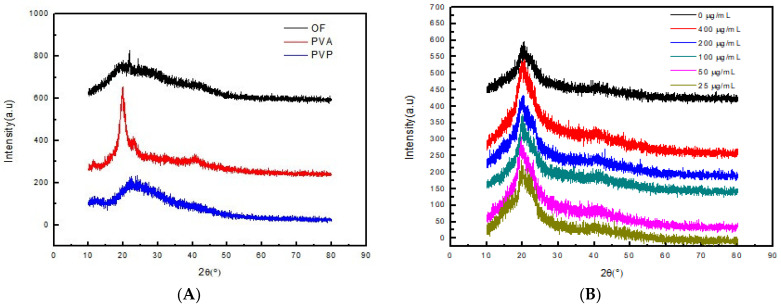
X-ray diffraction patterns of osmanthus extract, PVA, PVP and OF/PVP/PVA hydrogel (**A**): X-ray diffraction patterns of osmanthus extract, PVA, and PVP; (**B**): X-ray diffraction patterns of OF/PVP/PVA hydrogels with different concentrations of osmanthus extract.

**Figure 3 polymers-14-05399-f003:**
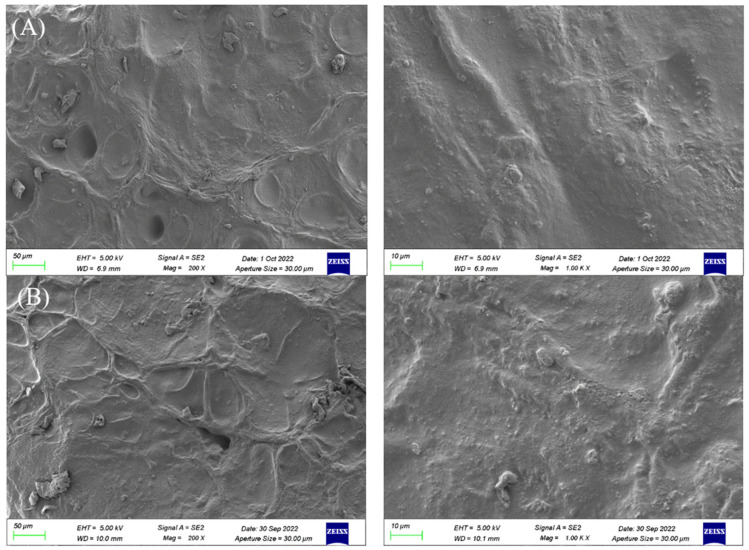
The microscopic morphology of PVP/PVA hydrogel and OF/PVP/PVA hydrogels with the concentration of 100 μg/mL under SEM. (**A**): SEM images of PVP/PVA (left: ×200; right: ×1000). (**B**): SEM images of OF/PVP/PVA hydrogel with a concentration of 100 μg/mL (left: ×200; right: ×1000).

**Figure 4 polymers-14-05399-f004:**
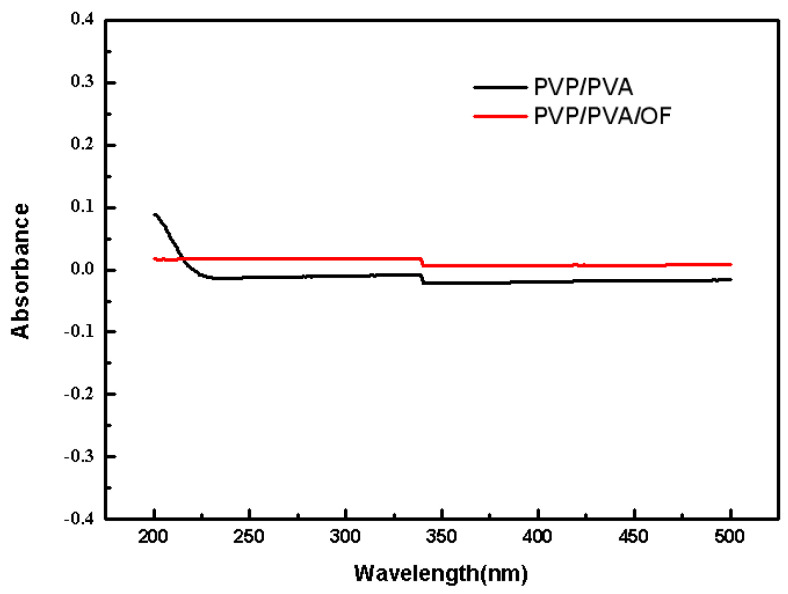
The stability of PVP/PVA hydrogel and OF/PVP/PVA hydrogels with a concentration of 100 μg/mL.

**Figure 5 polymers-14-05399-f005:**
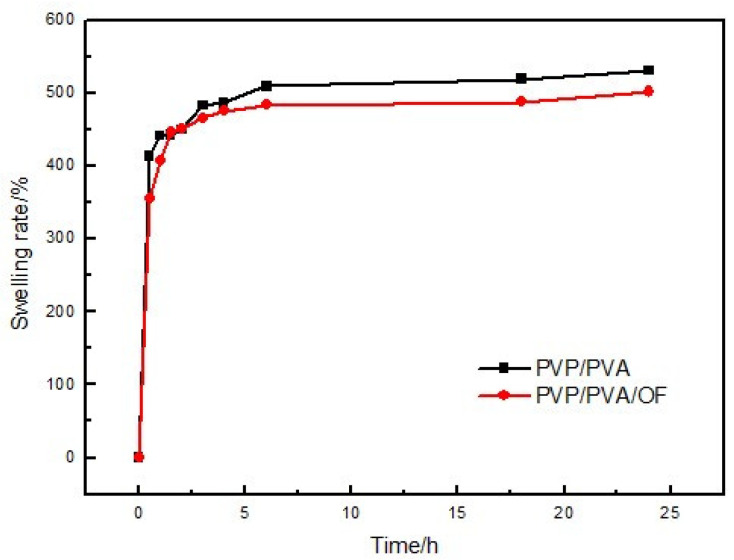
The swelling rate of PVP/PVA hydrogel and OF/PVP/PVA hydrogels with a concentration of 100 μg/mL.

**Figure 6 polymers-14-05399-f006:**
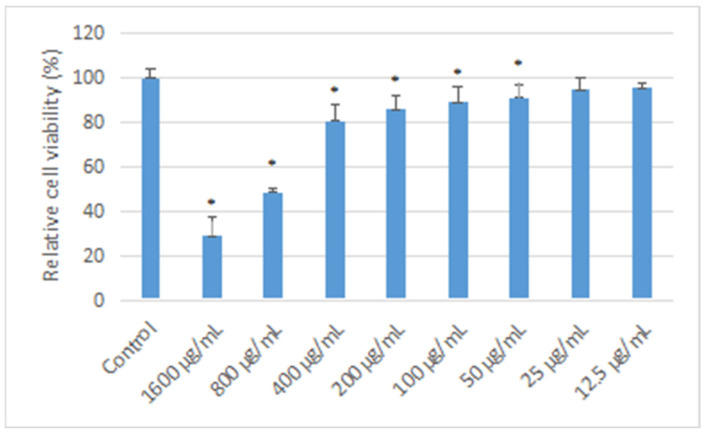
Effect of OF/PVP/PVA hydrogels on the cell viability of CAL-27 OF/PVP/PVA hydrogels with different concentrations of osmanthus extract were co-cultured with CAL-27 cells for 24 h, and cell viability was detected by MTT assay. *: *p* < 0.05 compared with control group.

**Figure 7 polymers-14-05399-f007:**
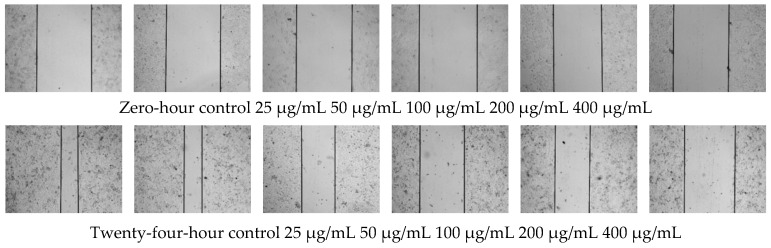
Effect of OF/PVP/PVA hydrogels on migration of CAL-27 cell scratching assay was performed. The area of scratch was observed under a microscope (×10).

**Figure 8 polymers-14-05399-f008:**
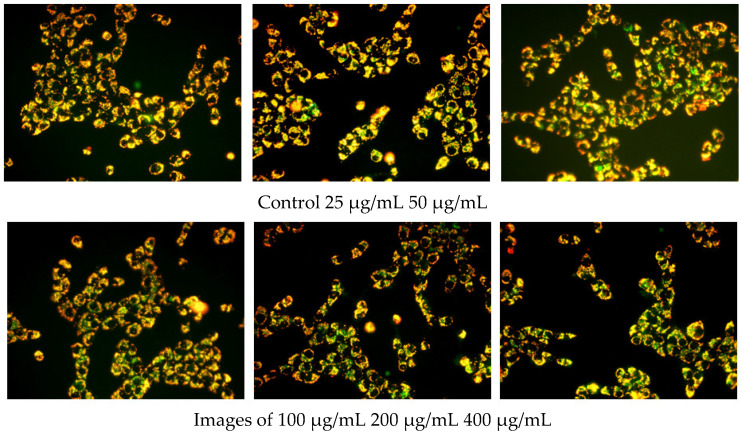
Effect of OF/PVP/PVA hydrogels on mitochondrial membrane potential of CAL-27 JC-1 staining was performed. The fluorescence was observed under a microscope (×40).

**Figure 9 polymers-14-05399-f009:**
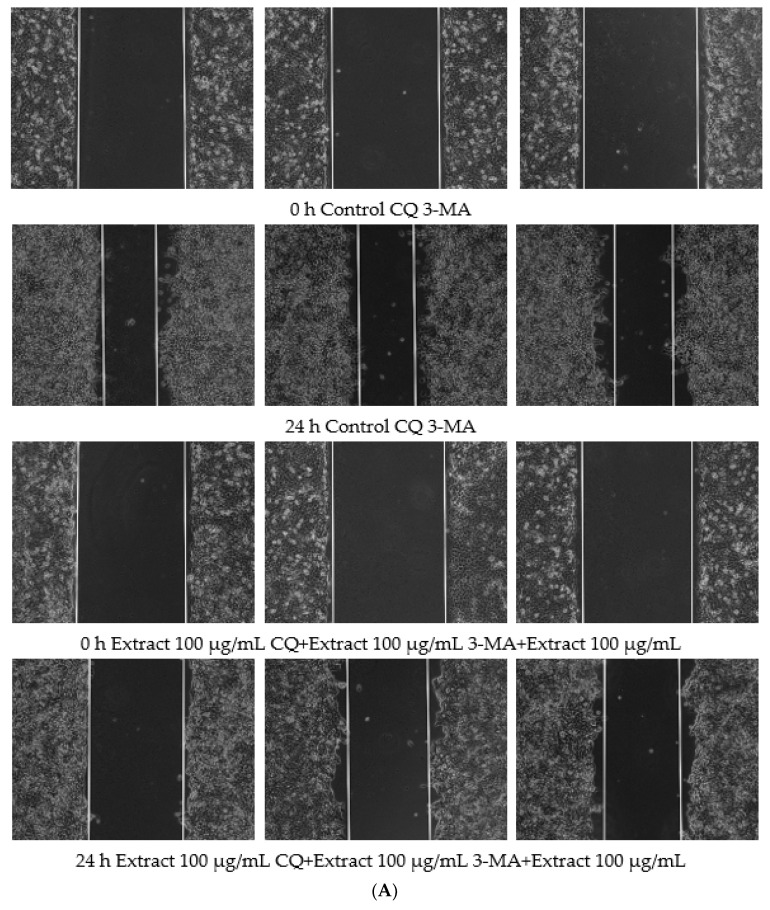
CQ and 3-MA attenuates the inhibitory effect of OF/PVP/PVA hydrogels by activating autophagy of Cal-27 cells. A: scratching assay; B: JC-1 staining.

## Data Availability

Not applicable.

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
