# Peer review of "Osmanthus-Loaded PVP/PVA Hydrogel Inhibits the Proliferation and Migration of Oral Squamous Cell Carcinoma Cells CAL-27"

_polymers, 2022, doi:10.3390/polym14245399_

Round 1
Reviewer 1 Report
The manuscript claims the inhibitory effect of the synthesized osmanthus-loaded PVP/PVA hydrogels. My comments for the manuscript:
1. Recent studies about the topic must be included and cited in the introduction. Yhe background is unsufficient.
2. What do GO and KEGG stand for? It must be written in long form.
3. In part 2.5 "Mix 10% PVP and 8% PVA aqueous solution at a ratio of 1:2, and add osmanthus extract into the mixture to make the final concentration of 400, 200, 100, 50, 25 μg/mL, respectively." must be rewritten as 10% Pvp and ... were mixed...."
4. In part 2.8 "...the area of the scratches was measured by Image J software and cell migration rate was analyzed using the formula below." and the formula needs a number.
5. I did not understand anything from figures 1-4. They are unnecessarily too big, whereas the writings on the images are too small and can not be read. Actually this part of the study is very confusing, i suggest the authors to insert these images as a supporting information. The results and discussions part is very disordered and it is hard to focus on both these network part and other part with analyses such as FTIR, XRD.
6. In Figure 5, did the authors draw the 7 main components of osmanthus extract? If not, ireference should be added.
7. The scratch assay images are not with he same magnification. I find this assay insufficient, i suggest the authors to use a staining and analyze with confocal
8. Overall, i found 3.1, 3.21, 3.3 and 3.4 out of the scope of the experimental part. They are confusuing and is suggest them to be placed in supporting.
9. I could not match the explanation in 3.9 with the images in Figure 12. Can the authors give a more detailed explanation and also labeling for the fluorescence images in Figure 12.
10. The materials and instruments used are not present in experimental part. Please add brands and specifications.
Author Response
The attachment is our reply

Reviewer 2 Report
Some corrections are required for this manuscript to be accepted:
1. What are GO and KEGG? The meaning of these abbreviations is not found in the article.
2. In section 2.5., please add the molecular weight for the PVP and PVA and the manufacturer.
3. How do you prove that you obtained a hydrogel after irradiation and that the polymer mixture is insoluble in water/other solutions? The structure of the hydrogel was not investigated to determine the gel fraction, swelling degree, stability, and loading capacity.
4. In section 2.6., I recommend you add the full name of the instruments (FTIR, XRD, and SEM), and specify the resolution at which the FTIR spectra were acquired.
5. Please change the x and y scale in Figure 1A, it is not visible.
6. Please modify Figure 5B, the numerical data is written right on the errors.
7. ‘‘The infrared spectrogram OF OF/PVP/PVA is basically...’’ Please correct it.
8. In figure 7, the description is not written correctly. Replace A with B.
9. ‘‘Fig 8A and 8B are scanning electron microscopy images of PVP/PVA and PVP/PVA/OF...’’ Please correct it.
10. ‘‘Figure 9. Effect of OF/PVP/PVA hydrogels on the cell viability of CAL-27 OF/PVP/PVA hydrogels with different concentrations of osmanthus extract were co-cultured with CAL-27 cells for 24 h, and cell viability was detected by MTT assay. *: p < 0.05 compared with control group’’. It is not understood, please correct it.
11. Figure 10, to be arranged, the pictures depending on the concentration do not have the same size. Then, correct 400 µg/Ml with 400 µg/mL.
12. The descriptions for Figures 10 and 11 are very long and difficult to understand. I recommend you change them.
13. In the Conclusions section, I recommend that the authors highlight the significant outcome of this study and future perspectives for the following research.
14. There are many mistakes in the References, they should be reviewed carefully.
Author Response
- What are GO and KEGG? The meaning of these abbreviations is not found in the article.
GO and KEGG stand for Gene Ontology and Kyoto Encyclopedia of Genes and Genomes, respectively. We revised them in manuscript.
- In section 2.5., please add the molecular weight for the PVP and PVA and the manufacturer.
We added the molecular weight for the PVP and PVA and the manufacturer.
- How do you prove that you obtained a hydrogel after irradiation and that the polymer mixture is insoluble in water/other solutions? The structure of the hydrogel was not investigated to determine the gel fraction, swelling degree, stability, and loading capacity.
We added some experiments about the gel fraction, swelling degree, stability, and loading capacity in the manuscript.
- In section 2.6., I recommend you add the full name of the instruments (FTIR, XRD, and SEM), and specify the resolution at which the FTIR spectra were acquired.
We added the full name of the instruments and specified the resolution at which the FTIR spectra were acquired.
- Please change the x and y scale in Figure 1A, it is not visible.
We revised figure 1A.
- Please modify Figure 5B, the numerical data is written right on the errors.
We modified Figure 5B.
- ‘‘The infrared spectrogram OF OF/PVP/PVA is basically...’’ Please correct it.
We corrected it.
- In figure 7, the description is not written correctly. Replace A with B.
We switched the positions of pictures A and B.
- ‘‘Fig 8A and 8B are scanning electron microscopy images of PVP/PVA and PVP/PVA/OF...’’ Please correct it.
We corrected it.
- ‘‘Figure 9. Effect of OF/PVP/PVA hydrogels on the cell viability of CAL-27 OF/PVP/PVA hydrogels with different concentrations of osmanthus extract were co-cultured with CAL-27 cells for 24 h, and cell viability was detected by MTT assay. *: p < 0.05 compared with control group’’. It is not understood, please correct it.
We corrected it.
- Figure 10, to be arranged, the pictures depending on the concentration do not have the same size. Then, correct 400 µg/Ml with 400 µg/mL.
We corrected it.
- The descriptions for Figures 10 and 11 are very long and difficult to understand. I recommend you change them.
We changed them.
- In the Conclusions section, I recommend that the authors highlight the significant outcome of this study and future perspectives for the following research.
We highlighted the significant outcome of this study and future perspectives for the following research.
- There are many mistakes in the References, they should be reviewed carefully.
We revised the mistakes in the References.
Round 2
Reviewer 1 Report
Figures 1-5 must be transfered to SI.
All figures must be centered.
In Figure 8, the red labeling (A, B) is not visible can be replaced with white.
It is difficult to follow Figure 12. I suggest that the 0 h and 24 h images are laid side by side in one each line.
The figure labeling as A and B style is not identical for Figure 7, 8, 14, it must be singularized.
There are mistakes in the references, they must be revised and corrected.
Author Response
- Figures 1-5 must be transfered to SI.
We transfered Figures 1-5 to SI.
- All figures must be centered.
All figures have been centered.
- In Figure 8, the red labeling (A, B) is not visible can be replaced with white.
We changed the red labeling to white.
- It is difficult to follow Figure 12. I suggest that the 0 h and 24 h images are laid side by side in one each line.
The 0 h and 24 h images have been laid side by side in one each line.
- The figure labeling as A and B style is not identical for Figure 7, 8, 14, it must be singularized.
We unified the style of A and B in Figure 7, 8 and 14.
- There are mistakes in the references, they must be revised and corrected.
We corrected them.
Reviewer 2 Report
1. No changes were made to Figures 1A and 5B. Please correct them.
2. Not all references have been corrected. For example, reference 7: Lindemann A, Takahashi H, Patel AA, et al. Targeting the DNA Damage Response in OSCC with TP53 Mutations [J]. Journal of Dental Research, 2018, 97(6): 635-644. Next would be the references 8,9,10,11,12....
Author Response
- No changes were made to Figures 1A and 5B. Please correct them.
We corrected them.
- Not all references have been corrected. For example, reference 7: Lindemann A, Takahashi H, Patel AA, et al. Targeting the DNA Damage Response in OSCC with TP53 Mutations [J]. Journal of Dental Research, 2018, 97(6): 635-644. Next would be the references 8,9,10,11,12....
We corrected the references.